# A Study of Sensor Placement Optimization Problem for Guided Wave-Based Damage Detection [note 1]

**DOI:** 10.3390/s19081856

**Published:** 2019-04-18

**Authors:** Rohan Soman, Pawel Kudela, Kaleeswaran Balasubramaniam, Shishir Kumar Singh, Pawel Malinowski

**Affiliations:** Institute of Fluid Flow Machinery, Polish Academy of Science, 80-231 Gdansk, Poland; pk@imp.gda.pl (P.K.); kaleeswaranb@imp.gda.pl (K.B.); shishir.singh@imp.gda.pl (S.K.S.); pmalinowski@imp.gda.pl (P.M.)

**Keywords:** plate, guided waves, damage detection, optimization, sensor placement

## Abstract

Guided waves (GW) allow fast inspection of a large area and hence have attracted research interest from the structural health monitoring (SHM) community. Thus, GW-based SHM is ideal for thin structures such as plates, pipes, etc., and is finding applications in several fields like aerospace, automotive, wind energy, etc. The GW propagate along the surface of the sample and get reflected from discontinuities in the structure in the form of boundaries and damage. Through proper signal processing of the reflected waves based on their time of arrival, the damage can be detected and isolated. For complex structures, a higher number of sensors may be required, which increases the cost of the equipment, as well as the mass. Thus, there is an effort to reduce the number of sensors without compromising the quality of the monitoring achieved. It is of utmost importance that the entire structure can be investigated. Hence, it is necessary to optimize the locations of the sensors in order to maximize the coverage while limiting the number of sensors used. A genetic algorithm (GA)-based optimization strategy was proposed by the authors for use in a simple aluminum plate. This paper extends the optimization methodology for other shape plates and presents experimental, analytical, and numerical studies. The sensitivity studies have been carried out by changing the relative weights of the application demands and presented in the form of a Pareto front. The Pareto front allows comparison of the relative importance of the different application demands, and an appropriate choice can be made based on the information provided.

## 1. Introduction

Guided waves (GW) are one of the most promising tools for structural health monitoring (SHM) of large plate-like structures and have attracted a great deal of interest from the research community [1,2]. GW-based techniques work on the principle that that the waves will be reflected by discontinuities in the medium (boundaries, damage). By appropriately processing the responses and comparing with the baseline measurements, the damage can be both detected and isolated. The GW-based techniques allow fast inspection of large areas and, with an appropriate number of sensors, may be used not only for detection, but also for damage localization (through triangulation). It has been been shown that GW are sensitive to small levels of damage, as well as damage originating from several different mechanisms including impact, fatigue, as well as moisture- and temperature-induced damage scenarios [3,4,5,6,7,8]. The use of GWs in metallic components is quite common, and due to their success, they are finding several applications in different fields. For complex structures, the wave propagation phenomena is very complex [9,10], and in order to ensure good coverage and damage detection ability, a large number of sensors is required. The increase in the number of sensors is associated with an increase in cost and difficulties with the instrumentation of the system. Not only does the extra number of sensors increase the cost of implementation, but in applications like airplanes, the added sensors lead to added mass of the instrumentation, which leads to higher operating costs. Hence, there is a need to reduce the number of sensors to lower the cost of implementation, as well as the secondary costs without compromising on the quality of the SHM of the system [11]. For this purpose, optimization of the sensor placement is essential.

The problem of optimization is quite relevant and has started to attract attention from several researchers; unfortunately, the subject for SHM using GW has not been treated with enough rigor, and hence, this study addresses the lack of literature in this area. The earliest work in the area of optimization of sensor placement (OSP) made use of the analytical approach for determining the sensor placement [12,13]. Some recent work has built on this work and has proposed optimization taking into consideration the different parameters to which the GW-based SHM techniques are sensitive [14,15,16,17]. However, this approach is resource intensive and makes several simplifying assumptions. Thus, there is a need for a different approach to tackle the optimization problem. The present study aims at providing the framework of a methodology for the OSP through a combination of analytical, numerical, and experimental approaches. The combination of the three approaches promises to reduce the simplifying assumptions that lead to deviation between the computed optima and the reality, without allowing the effort required for optimization to increase. The methodology in the first stage is applied for a simple square aluminum plate and later to a triangular plate. It should be noted that the work presented is at the methodology level; it can be easily extended for complex structures with non-convex shapes, as well as anisotropic materials.

The paper is organized as follows. Section 2 presents the methodology framework; Section 3 formulates the optimization problem including the cost function and its analytical, numerical, and experimental implementation. Section 4 presents the equivalence and the complementary nature of the three methods through validation. The optimization results are presented in Section 5. Section 6 draws some conclusions and identifies the area of future work.

## 2. Methodology

The aim of the paper is to outline a sensor placement methodology based on analytical, numerical, and experimental analysis. The aim is to streamline the entire process and make it less resource intensive both in terms of computational and material resources. Thus, a combination of the three approaches is proposed to achieve accuracy at a reasonable cost. The information flow between the three approaches is shown in Figure 1.

Each of the three approaches has their own set of advantages and disadvantages, for instance the analytical approach allows faster computation and is less intensive in terms of computations, as well as requiring low levels of instrumentation. However, the simplifying assumptions made for the analytical approach make it too simplistic, and it does not reflect the true nature of the problem. In the numerical approach, with the development of novel modeling approaches and parallel computing, the time requirement for the computations is lowered considerably, but this approach remains computationally demanding, requiring several days for simulated studies. Furthermore, the numerical modeling uses some simplifying assumptions, and the results obtained are often too idealistic as compared to real applications. Furthermore, the numerical approach has limited scalability, as a minor difference in the geometry of the sample necessitates repetition of the entire simulation process. The experimental approach is the most time consuming and resource intensive of all. It requires time for preparing the samples, performing the measurements, and analyzing the data. The experimental approach is not scalable or customizable for other application of the sensor placement. Furthermore, the experimental campaign is not always reversible as the sensors once placed may not be easily removed, thus affecting the wave propagation irreversibly. However, the experimental approach reflects all the facets of the optimization problem, as well as the uncertainties in the measurements. It also takes into consideration all the physical phenomena occurring in the sample, some of which may not be possible to replicate in the numerical and analytical approach. Hence, by using a combination of the three approaches, the individual shortcomings can be overcome while allowing synergy in the advantages offered by each of the approaches.

## 3. Optimization

The general framework of the OSP problem was given by Ostachowicz et al. [18] as shown in Figure 2.

According to the framework, the first step is the definition of the application demands. As the OSP deals with the use of GW for SHM, the application requirements identified are:(i)Maximum coverage of the structure with at least one sensor-actuator pair (coverage1)(ii)Maximum coverage of the structure with three sensor-actuator pairs (coverage3)(iii)Minimum number of sensors (*s*)

The coverage1 is the area covered by at least one sensor actuator-pair as a result of which damage may be detected at these locations, but may not be isolated adequately. The coverage3 is the area covered by at least three sensor-actuator pairs. For an isotropic plate, the area covered by a sensor-actuator pair has an elliptical shape. Each pair of sensor-actuator defines an ellipse with the sensor and actuator as the foci. The intersection of the three or more ellipses (from three sensor-actuator pairs) allows the isolation of the damage. The number of sensors (*s*) relates to the number of deployed sensors, which need to be low to reduce the associated instrumentation and secondary costs.

For the SHM using GW, the use of piezo-electric transducers is an industry standard for the active SHM of plate-like structure. The circular piezo sensors offer a symmetric wavefront in the isotropic medium and hence are preferred. The operational parameters chosen for the optimization were the minimum distance between the sensors based on the outer diameter of the sample and minimal distance from the edges of the sample. Based on the experience of the authors, the distance from the edges and candidate locations for sensor placement was determined as 10-times the outer diameter of the sensors used. The distance from the edges allows easier distinction of the reflected wave and the incident wave and hence was chosen. The grid size of the candidate locations affects the problem size. A rarer mesh for the grid leads to a smaller problem size, but it also may lead to a sub-optimal solution. On the other hand, a denser mesh allows a thorough search in the sample space, but the performance of the optimization algorithm suffers, while the computation time goes up exponentially. Thus, the grid size of 10 × 10 was chosen as a trade-off between the thoroughness of the search space while limiting the computation time.

Based on the application demands, there is more than one objective for optimization, and as a result, multi-objective optimization is necessary. Due to the complexity of the problem at hand and to allow application of a wide range of optimization tools, scalarization [19] of the cost function is necessary. The scalarization is achieved by assigning different weighing factors to the different objectives and then combining the objectives to yield a single-valued cost function. The cost function for the application at hand based on the three application demands is given by Equation (Equation 1):(1)$$cost=-1\times (\beta \;\frac{coverage3}{s^{\gamma }}+\left(1-\beta \right)\;\frac{coverage1}{s^{\delta }})$$
where coverage3 is the % of points of the grid that lie within the sensing range of three or more sensor-actuator pairs and coverage1 is the % of points that lie in the sensing range of a single sensor-actuator pair. β, γ, and δ are weighting values to determine the relative merit for each of the parameters, and *s* is the number of sensors. The parameters γ and δ can be treated as independent of each other or dependent based on the choice. The two parameters were introduced to show the different correlations of the coverage3 and coverage1 values to the number of sensors (*s*).

It is worth mentioning that the scalarized cost function is in no way unique and for specific applications may be modified to suit the requirements. The cost function should take into consideration all the application demands. Care should be taken that the objectives are scaled appropriately so that the cost function is not overly dominated by a single objective. It is common in optimization problems that the scalarized cost function lacks physical significance. However, the cost function and its performance may be linked to some quantifiable metric such as accuracy of damage detection and isolation or the probability of detection of damage, etc. The definition of the cost function goes a long way in determining the suitable algorithm for the optimization, as well as the computational demands for optimizing.

Once the cost function is determined, based on the scalarized cost function, which is not linear, continuous, and differentiable, it is evident that the meta-heuristic algorithms are suitable for this class of optimization problems. The sequential placement optimization techniques need deterministic computations and hence are computationally efficient, but meta-heuristic techniques outperform them in terms of obtaining the optimal solution [18] and hence were considered. Several meta-heuristic algorithms such as the particle swarm optimization, ant-colony optimization, GA, etc., for the OSP problem have been proposed and the advantages and disadvantages discussed by Ostachowicz et al. [18]. According to the “no free lunch algorithm” [20], the mean performance of all the optimization techniques over all the classes of problems is similar. Any improvement in the performance for a certain class of problems will be reflected by a drop in the performance in other class of problems. Few studies comparing the performance of different algorithms for a given class of problem have been performed [21,22]. However, the objective comparison of two or more optimization algorithms is difficult as their performance depends on the definition of the cost function, the formulation of the optimization problem, as well as the fine tuning of the key parameters for the implementation of the algorithm. As a result, the comparative study is considered outside the scope of this paper.

Based on the simplicity of the implementation, for the large problem size, and its popularity in similar applications [23,24], the genetic algorithm (GA) was chosen as the optimization algorithm. The flowchart for the implementation of GA is shown in Figure 3.

The integer GA was implemented with each chromosome depicting a sensor network configuration and each gene of the chromosome depicting a sensor location. For the computation of the cost function, the analytical approach was necessary, while for the provision of inputs for the analytical approach, the numerical and experimental approaches were used.

### 3.1. Numerical Approach

The numerical approach is necessary for giving insight into the complex wave interactions that occur. The key aspect for using signal processing tools for damage detection is the estimation of the first arrivals of the waves and the ability to differentiate between the arrivals of the reflections from structural components and damage. The numerical model allows us to visualize the wave propagation in time and space, which is possible with the experimental approach only in a few conditions. Furthermore, at a later stage when more structural features like stiffeners and rivets will be investigated, the knowledge of these physical interactions will allow updating the analytical approach for the complex cases. The numerical model provides the wavefront for the propagation.

In the first case, a plate of 1 m × 1 m in size and a thickness of 1 mm was investigated (shown in Figure 4).

The material properties of aluminum alloy were assumed as follows: Young modulus E = 72 GPa, Poisson ratio ν = 0.33, mass density ρ = 2660 kg/m${}^{3}$. The material damping was not incorporated into the model.

The time domain spectral element method was employed using an in-house-developed and validated code for the problem of Lamb wave propagation phenomenon [26,27]. 3D solid spectral elements were used for modeling the analyzed plates with bonded piezoelectric transducers. Parallel implementation on graphics processing unit (GPU) was used to reduce the computation time [28]. It was assumed that the piezoelectric transducers were perfectly bonded to the surface of the plate (the bonding layer was not modeled). A piezoelectric transducer in the form of a disc of a diameter of 10 mm and a thickness of 0.25 mm was modeled by 3D spectral elements. Classic electromechanical coupling was assumed. The analysis was performed for the excitation signal in the form of a tone burst of carrier frequency $f_{c}$ = 100 kHz and modulation frequency $f_{m}$ = 20 kHz. The selected spectral elements for modeling the host structure, as well as piezoelectric transducers had 36 in-plane nodes and three nodes across the thickness, as shown in Figure 5a.

The wave propagation patterns (frames) in the form of out-of-plane (transverse) displacements at three selected time instances for Transducers 1–3 working as actuators are presented in Figure 5b.

As can be seen, the wave propagation front is circular due to the isotropic material. Furthermore, it was seen that for out-of plane measurements, A0 mode was dominant, and as a result, the signal processing for damage identification was easier and hence was chosen for the time of flight calculations. It is noted that the time for the simulation using a CPU: Intel I5 4460 3.2 GHz, 16 GB RAM, was 215 h, while the same using a GPU: Nvidia Tesla K20X 6 GB RAM, was 36 h. Hence, although the GPU reduced the computation time, the time requirement was very large, and hence, analytical implementation of the optimization problem was necessary.

### 3.2. Experimental Approach

As mentioned above, the damping was not considered in the numerical approach. In order to obtain the attenuation values, an experimental approach is necessary. Furthermore, the experimental approach allows the study of the physics of the interaction between the waves and the different structural features like stiffeners, rivets, etc. Furthermore, the ultimate goal of the sensor placement optimization is the deployment of the sensors, the experimental validation of the sensor placement, and the ability to detect damage. Hence, damage will be simulated at different locations, and the ability of the sensor network to detect the damage will be investigated in the future. The performance will be compared to other placement strategies to determine the performance of the sensor placement optimization exercise.

At some times, when the properties of the material or the constitution is not exactly known, the experimental approach may be necessary for obtaining the directionality of the wave propagation and the validation of the numerical model. Figure 6 shows the experimental specimen and the sensor placement similar to the numerical model.

The measurements were carried out using the PZT sensors and actuators in the pitch catch mode. In order to apply and register electric voltage to and from the transducers, an integrated generation/acquisition system was used. It had 13 channels, 12 of which can serve for acquisition and one for generation at any moment. The system was connected to a personal computer with a USB cable and controlled by an application implemented in the MATLAB^®^ environment. The MATLAB environment allows us to choose the number of channels to be used, the number of cycles in the excitation (sine), the frequency of excitation, the modulation type (Hanning, triangle, or rectangular window), and which channel is a wave generator. 200 kHz was chosen as the frequency of excitation, and the sampling frequency was 1.25 MHz. The PZT sensors used for the actuation and sensing were manufactured by Ceramtec out of Sonox P502 material. These transducers were monolithic piezoelectric discs with a diameter of 10 mm and a thickness of 0.5 mm.

The PZTs allow measurements only at specific points, so in addition to the PZTs, measurements were also carried out with scanning laser Doppler vibrometer (LDV) (Polytec 3D Laser Scanning Vibrometer PSV400). The LDV allows visualization of the wave propagation and the interaction with the edges and other discontinuities in the structure. The measurements were carried out in the 1D mode (only one scanning head used). In this setup, only out-of-plane components of the displacements were measured. The actuation was achieved by the same PZT actuators. The actuation from an external generator was amplified and synchronized with LDV sensing.

### 3.3. Analytical Approach

As shown in the numerical simulation section, the cost for the simulation of the different possibilities of sensor placements is prohibitively high, and as such, the analytical approach is recommended. For the isotropic aluminum case, the wave velocity in all directions was considered constant. Furthermore, aluminum is a well-studied material, and the wave velocities have been well documented. Furthermore, the attenuation in aluminum is quite low, so the range was considered large enough for the plate under investigation. Therefore, the scope of the tasks under the analytical approach were to determine the highest coverage obtained by a particular sensor-actuator pair. This can be achieved by determining the largest ellipse that will fit in the plate. This was done by solving the equation for tangency of ellipse with known foci (sensor-actuator pair locations). This yields three conditions for determining the unique largest ellipse fitting inside the plate boundaries. The equation for the major axis and minor axis for the ellipse tangent to the edges of the plate are given by Equations (Equation 2) and (Equation 3).
(2)$$\begin{array}{c} a=[lcx^{2}\cdot sA^{4}+2\cdot cA\cdot M\cdot sA\cdot sx+M\cdot sA^{2}\cdot sx^{2}+sA^{4}\cdot sx^{2}\cdot xc^{2}+lcx\cdot sA^{4}\left(2\cdot sx\cdot xc-2\cdot yc\right)\\ -2\cdot sA^{4}\cdot sx\cdot xc\cdot yc+sA^{4}\cdot yc^{2}+cA^{4}\cdot (lcx^{2}+2\cdot lcx\cdot sx\cdot xc+sx^{2}\cdot xc^{2}-2\cdot lcx\cdot yc\\ -2\cdot sx\cdot xc\cdot yc+yc^{2})+cA^{2}(M+sA^{2}(2\cdot lcx^{2}+4\cdot lcx\cdot sx\cdot xc+2\cdot sx^{2}\cdot xc^{2}\\ -4\cdot lcx\cdot yc-4\cdot sx\cdot xc\cdot yc+2\cdot yc^{2}{))]}^{1/2}/{\left[\left(cA^{2}+sA^{2}\right)\left(1+sx^{2}\right)\right]}^{1/2} \end{array}$$
(3)$$\begin{array}{c} b=[-M\cdot sA^{2}+2\cdot cA\cdot M\cdot sA\cdot sx+cA^{4}(lcx^{2}+2\cdot lcx\cdot sx\cdot xc+sx^{2}\cdot xc^{2}-2\cdot lcx\cdot yc\\ -2\cdot sx\cdot yc+yc^{2})+sA^{4}\left(lcx^{2}+2\cdot lcx\cdot sx\cdot xc+sx^{2}\cdot xc^{2}-2\cdot lcx\cdot yc-2\cdot sx\cdot xc\cdot yc+yc^{2}\right)+\\ cA^{2}(2\cdot lcx^{2}\cdot sA^{2}-1\cdot M\cdot sx^{2}+lcx\cdot sA^{2}\left(4\cdot sx\cdot xc-4\cdot yc\right)+\\ sA^{2}\left(2\cdot sx^{2}\cdot xc^{2}-4\cdot sx\cdot xc\cdot yc+2\cdot yc^{2}\right){)]}^{1/2}/{\left[\left(cA^{2}+sA^{2}\right)\left(1+sx^{2}\right)\right]}^{1/2} \end{array}$$
where sA is the sine of alpha, cA is the cos of alpha, lcx is the *x* intercept of the tangent line, sx is the slope of the line, *M* is the square of the distance between the sensor and actuator, xc is the x co-ordinate of the center of the ellipse, yc is the y co-ordinate of the center of the ellipse, and *a* and *b* are the semi minor and major axes of the ellipse. The nomenclature and the approach are concisely explained in Figure 7a

Once the ellipse equation was analytically computed for all the edges of the plate, the minimum values for *a* and *b* each were taken as the largest ellipse inscribed in the plate. The GW-based damage detection techniques extract the damage scatter features from the complex signal through comparison with the baseline [29]. The direct arrival wave and the TOF are key features required for damage localization with these techniques. The boundary reflected damage scatter features were not present in the healthy signal, and hence, they were not accounted for in the baseline comparison. This may lead to incorrect identification of damage. Hence, for the proper functioning of the damage detection algorithm and to limit the boundary reflections, the largest ellipse inscribed completely in the structure is taken as the maximum coverage area for the sensor-actuator pair. For more advanced signal processing techniques that are able to account for the reflected damage scatter, the ellipse region may be increased by a factor, as shown by Salmanpour et al. [15]

The use of the tangency condition is required as the ability of the basic signal processing algorithms for location and detection of damage is compromised for cases where the wall reflections occur. Each of the points lying inside the ellipse can be investigated using the sensor pair. The process was repeated for each sensor pair and superimposed in order to obtain the coverage for the sensor network. For the detection of damage, it is sufficient that a point is investigated by one sensor-actuator pair (coverage1), but accurate damage isolation was carried out using the triangulation technique, which requires that a given point needs to be investigated by at least three sensor-actuator pairs (coverage3). Figure 7b explains the nomenclature for coverage1 and coverage3. It should be noted that the intersection of two ellipses does not allow damage isolation and as such does not add any significant information, so is not considered as an objective for optimization.

## 4. Validation

To verify if the largest ellipse fitting inside the plate was computed correctly, the results obtained from the time of flight analysis for the numerical model and the experimental analysis were compared with the analytical approach. The signals from the numerical model and the experimental data have been reported in [11] and are in reasonable agreement. Figure 8 shows the A-scan for the numerical simulation. Based on the full-field simulations, the arrivals of the reflections can be easily identified. The full-field wave forms are shown in Figure 9.

The transformation scheme for obtaining the ellipses from the A-scans has been explained in Figure 10. The time window taken for the energy calculation was half the time period of one excitation cycle. The time interval after the arrival of the direct path was considered for the energy calculation as the focus of the elliptical approach was on the waves reflected from the edges or damage. The ellipses corresponding to the maximum energy calculated through the process in Figure 10c for the A-scans from the numerical and experimental approach are shown in Figure 11 and compared with the ellipses obtained analytically. As can be seen, the ellipses obtained from the three approaches are in exact agreement and hence pointed to the complementary nature and equivalence of the three approaches.

A similar transformation scheme was followed for the A-scans obtained from the LDV measurements, and the ellipses corresponding to different time instance are shown in Figure 12. As can be seen, the innermost ellipse was based on the reflection from the mass placed at the “x” mark. The red ellipse is based on the reflection from the left boundary of the plate. The light blue ellipse passes through the second “x” mark where another mass was placed. In the signal processing algorithm used for the purpose of the study, if the time signals after the first reflection from the boundary were discarded, the reflection from the second mass would be discarded, and as a result, the sensor pair 4–5 will not cover the location of the second mass. On the other hand, the first mass will be detected by the sensors at Locations 4 and 5. This shows that the largest ellipse approach is valid based on numerical and experimental results and may be used for the OSP.

## 5. Optimization Results

### 5.1. Square Plate

For the coding of the GA, each gene of the chromosome corresponds to the sensor position. The sensor positions assigned for different integers are shown in Figure 13. The GA iteratively generates sensor placement configurations and assesses their suitability based on the cost function chosen. The sensor configurations are then sorted based on their suitability. Fifty percent of the sensor networks (chromosomes) are retained (elitism), while the others are discarded. The discarded networks are replaced by the new generation, which is formed by mating of the retained chromosomes. Some degree of mutation (1%) in the retained population is induced for conducting a thorough search in the close vicinity of the retained population. The number of generations refers to the number of iterations of the selection, mating, and mutation (shown in Figure 3) that are allowed for the search of the optimal solution. The GA is a meta-heuristic optimization algorithm and for large problem size does not guarantee complete convergence. Hence, several runs of the GA were made, which yielded different sensor placements. However, each of the sensor placement yielded better results than the random placement, as shown in Table 1. It should be noted that the GA was run for minimizing the cost function identified in Equation (Equation 1). However, only the coverage3 and coverage1 objectives were represented, as the cost function lacks physical significance due to the scalarization.

It can be seen that for each of the GA runs, the sensor network obtained was different, and this points to the fact that convergence had not been achieved even after 5000 generations. However, the improvement in the solution even after 10,000 generations was marginal and hence did not justify the additional computational effort; hence, 5000 generations were chosen as the stopping criterion. Convergence was not achieved as the optimization problem size was large: there were ∑i=08181i
$=2.41\times 10^{24}$ possible sensor placements. It should be noted that for the evenly-placed sensors, the performance for the coverage3 metric was better, while that for the coverage1 metric was worse than that for the placement obtained from the GA for the same number of sensors (Run 3). Therefore, the relative weightings of the two metrics will affect the optimal solution. In addition, although the even placement is intuitive for a simple isotropic plate, this is not the case for complex shapes or for anisotropic materials, and hence, there is a need for optimization.

Based on the sensor placements for Run 3 and Run 5 (shown in Figure 13), it was evident that the GA was converging towards the sensor placement along the diagonals of the square plate. Therefore, the coverage statistics for the sensors placed along the diagonals was investigated. The coverage obtained was superior to those achieved by the GA runs. However, the fact that the GA was slowly converging towards the diagonal placement points towards the fact that the GA algorithm indeed had been satisfactorily coded and developed. Furthermore, it should be stressed that placement along the diagonals even though quite intuitive is not always the case for anisotropic materials or plates with structural features. Hence, there is merit in the use of optimization techniques for OSP.

The coverage plots for Run 3 and the random placement are shown in Figure 14. The color map indicates the locations along the plate with coverage achieved by different numbers of sensor-actuator pairs. The locations with dark blue (towards the corners) had no coverage, while the yellow-colored areas had coverage with a large number of sensor-actuator pairs. A higher coverage (>3) in one particular position is redundant and does not add much value to the assessment of the structure. The optimal placement showed a more even spread of the coverage, while the random placement showed a higher coverage in the central part of the plate, which was also seen in several other non-optimal placements. Therefore, the key to achieving a good coverage is a wide distribution of the sensors.

### 5.2. Triangular Plate

The problem size for the square plate was very large, and as a result, convergence was not achieved with the GA. In order to check the effectiveness of the GA for finding the optimal placement, as well as to show the flexibility of the methodology, the OSP problem was posed for a triangular plate, as shown in the Figure 15a. The reduced size of the plate gives the possibility of fewer sensor placements given by ∑i=04949i$=5.62\times 10^{14}$, which can be investigated with brute-force. Thus, the triangular plate sample was used for the assessment of the performance of the GA and to check if indeed the GA developed can achieve convergence to the optimal solution.

The parameters of the GA were chosen as the number of generations = 5000 and the number of chromosomes = 64. The mutation rate = 1%, and elitism = 50%. The cost function was chosen similar to the previous simulation for the square plate. The coverage for the optimized placement is shown in Figure 15b. The optimization algorithm indeed yielded the optimal result and served as a validation of the algorithm. Again, the optimized sensor placement was symmetric, but this was only true for the isotropic plate; hence, there is merit in optimization for complex plates, which is identified as work for the future.

### 5.3. Effect of Scalarization

The multi-objective optimization was simplified through scalarization. In the present application, there were three metrics that needed to be optimized. The coverage3 and coverage1 are complementary in the sense that both need to be maximized, while the number of sensors needs to be minimized.

In order to study the effect of different objectives and their weighting factors, sensitivity studies were run on a problem of reduced size. The candidate locations for the sensor were reduced to 16 (instead of 81). For the reduced problem size, all possible sensor placements were analyzed. Figure 16 shows the maximum coverage and the mean coverage achieved by the network for the chosen number of sensors.

As can be seen, the the mean coverage increased as the number of sensors grew, while the maximum achievable coverage increased initially and then converged to the maximum. While using the GA for the optimization especially for a large problem size, absolute convergence to the global optima is not guaranteed, and as such, the optimal solution obtained will lie between the mean coverage and the maximum coverage. At the same time, for a large problem size, the mean coverage and maximum coverage for a given number of sensors are not known a priori. For such cases, the methodology proposed by Croxford et al. [31] is useful. In the paper, the authors developed a method for determining the minimum number of sensors per unit area required for reliable damage detection under environmental uncertainties. Based on a similar study, the minimum number of sensors may be determined and used as a constraint in the optimization. The plot also supports the idea of using two parameters γ and δ in the cost function as the correlation of the coverage3 and coverage1 was different to the number of sensors. The coverage1 value was more independent of the number of sensors as compared to the coverage3 metric, especially when increasing the number of sensors used. It should also be noted that even in the case for γ=δ, the optimal placements obtained for maximizing the coverage1 and coverage3 metrics were significantly different.

The effect of the value of β on the two metrics (coverage1 and coverage3) is shown in Figure 17. To show the relative importance of the two objectives alone, the values of γ and δ were set to zero.

As can be seen, for an increasing number of sensors, the coverage3 and coverage1 values increased. Furthermore, it can be seen that for different values of β, different sensor placements were obtained, allowing different coverage1 and coverage3 values. The decision on the value of β may be determined based on the threshold value of the coverage values required for the application at hand. For instance, for critical applications, it might be necessary to detect damage while the isolation of damage may be carried out through other NDEand NDTtechniques, in this case, a lower value of β may be necessary. Furthermore, in some other non-critical applications, the isolation of damage may be just as important as the detection. The Pareto front allows us to make an informed decision based on the options at hand.

## 6. Conclusions

The paper outlines a methodology for OSP for damage detection and isolation using the GW technique. The methodology is based on a combination of analytical, numerical, as well as experimental approaches where the input for the optimization is secured using experimental data and the insights into the complexity of the problem are gained from the numerical simulations.

The simplified ellipse approach has been shown to be equivalent and physically relevant through numerical and experimental results. It has also been shown that the methodology yields an improved coverage using the same number of sensors, faster computation speeds as compared to the numerical only approach or experimental only approach, as well as lesser simplifying assumptions than purely a numerical or analytical approach. The better coverage ensures a more reliable SHM of the specimen. The optimization approach was validated on a problem of reduced order size, and the optimization did indeed give a convergent optimal solution for the chosen weights of the contradicting objectives.

The paper also presents sensitivity studies and the effect of scalarization on the optimization problem. The effect of changing the number of sensors and the relative weights of the coverage3 and coverage1 metric were shown in the form of the Pareto front. The Pareto front allows the decision-makers to make an informed decision on the placement of the sensors based on the desired values of the different objectives. This can allow a tailored placement based on the application demands or the availability of the sensors.

Although the present methodology has been applied to simple structures, this method can be easily extended to anisotropic materials and complex geometries [14,32]. For anisotropic materials, the parametric shape for wave propagation may be obtained from numerical simulations or the experimental approach, and the tangency condition for the shape may be determined. The ray-tracing approach may be used for anisotropic materials, as well [33,34]. If the ray-tracing approach is employed, only the determination of the coverage area of a sensor-actuator will change; all other steps of the methodology would remain unaffected.

The above outlined methodology has the potential to be applied to industrial applications where plate-like structures such as airplane wings components, etc., are used. The geometry of components typically used in industrial applications is more complex and has several structural features such as rivets, stiffeners, holes, etc., and investigation into their presence is identified as the next step for research. Furthermore, the study of the use of this technique for anisotropic materials to ensure a wider range of applications will be undertaken. Once the methodology is shown to be robust for these complexities, the application of the methodology for real structures will be undertaken.

## Figures and Tables

**Figure 1 sensors-19-01856-f001:**
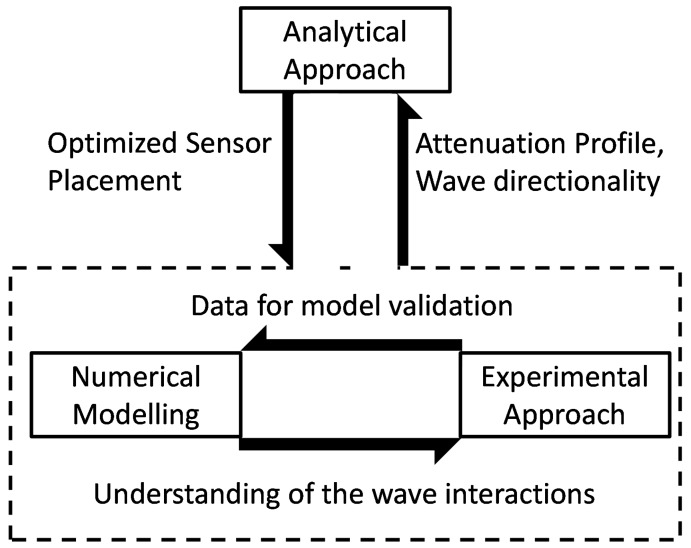
Synergy and data transfer between three approaches [11].

**Figure 2 sensors-19-01856-f002:**
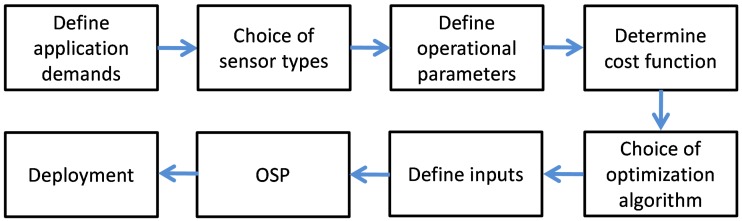
Framework of the optimization of sensor placement (OSP) problem [18].

**Figure 3 sensors-19-01856-f003:**
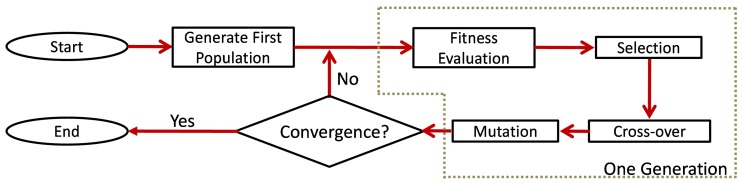
Framework of the simple genetic algorithm (GA) [25].

**Figure 4 sensors-19-01856-f004:**
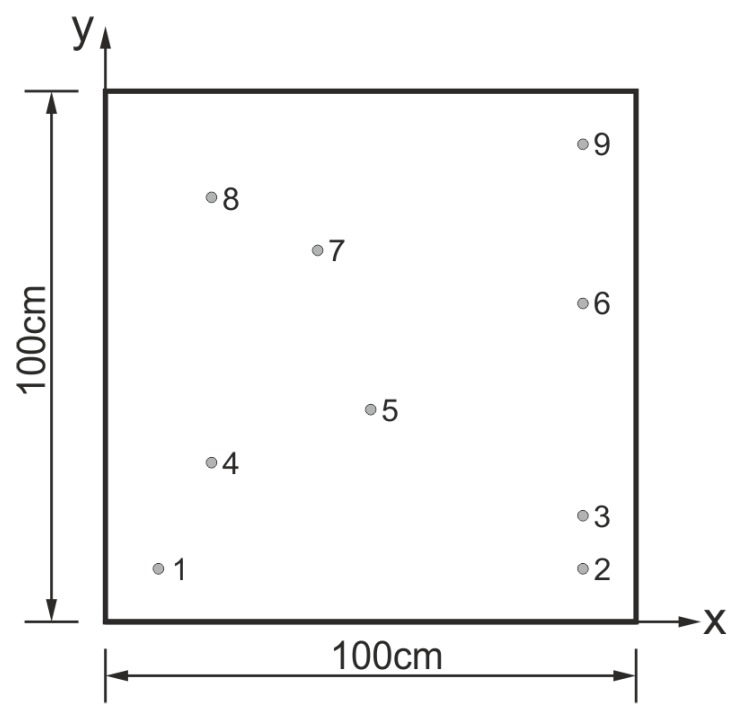
Geometry and arrangement of piezoelectric transducers on a square aluminum plate.

**Figure 5 sensors-19-01856-f005:**
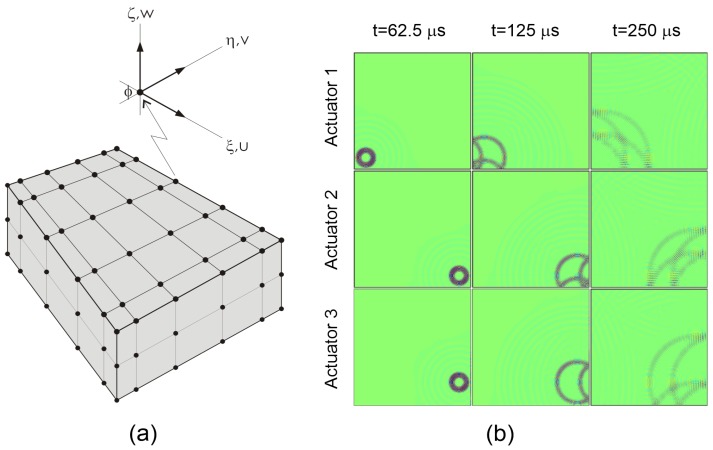
(**a**) Modeling approach showing 3D spectral element with three nodes across the thickness; (**b**) Wave propagation patterns at selected times propagating from Actuator Nos. 1, 2, and 3, respectively.

**Figure 6 sensors-19-01856-f006:**
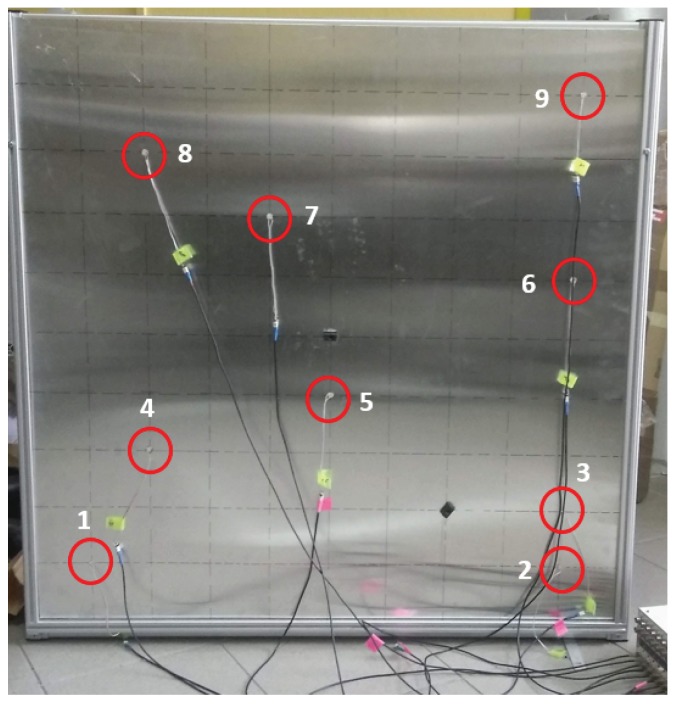
Aluminum plate with PZT sensors.

**Figure 7 sensors-19-01856-f007:**
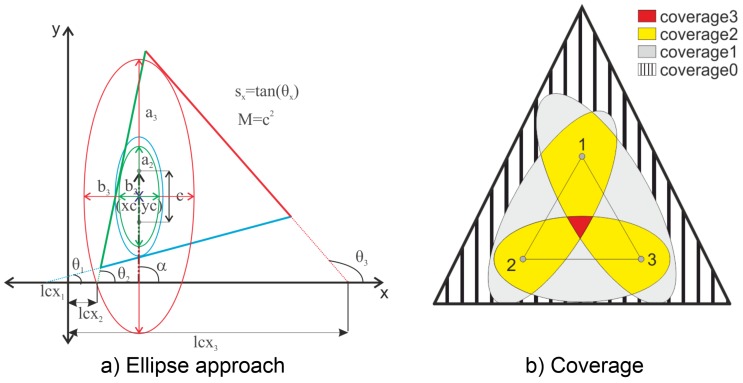
(**a**) Nomenclature for the ellipse approach for a triangular plate; (**b**) Explanation of coverage terminology.

**Figure 8 sensors-19-01856-f008:**
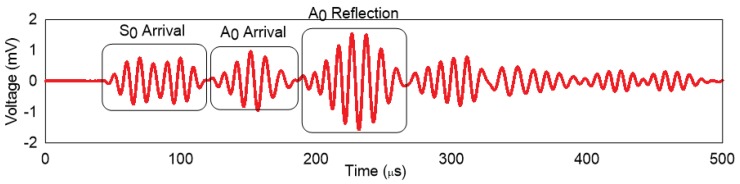
Wave propagation for the numerical model (Actuator-1, Sensor-4).

**Figure 9 sensors-19-01856-f009:**
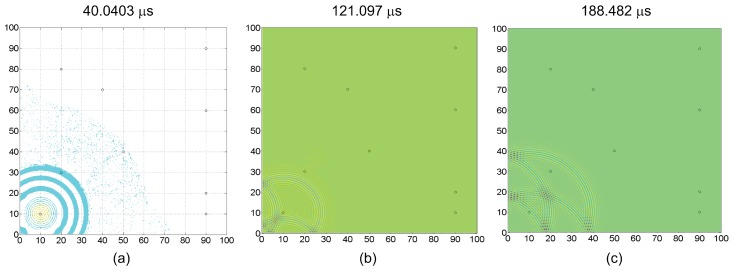
Wave propagation for the numerical model with actuation at 1 and sensing at 4. (**a**) Arrival of the S0 wave; (**b**) Arrival of the A0 wave; (**c**) Arrival of the reflected A0 wave [11].

**Figure 10 sensors-19-01856-f010:**
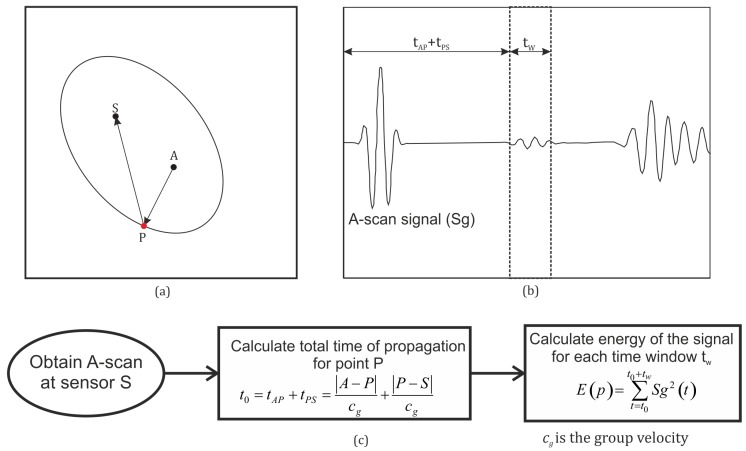
Scheme for obtaining ellipses from the A-scans. (**a**) Plate with the actuator (A), sensor (S), and point (P) on the plate; (**b**) A-scan at sensor S; (**c**) Flowchart for obtaining the ellipse.

**Figure 11 sensors-19-01856-f011:**
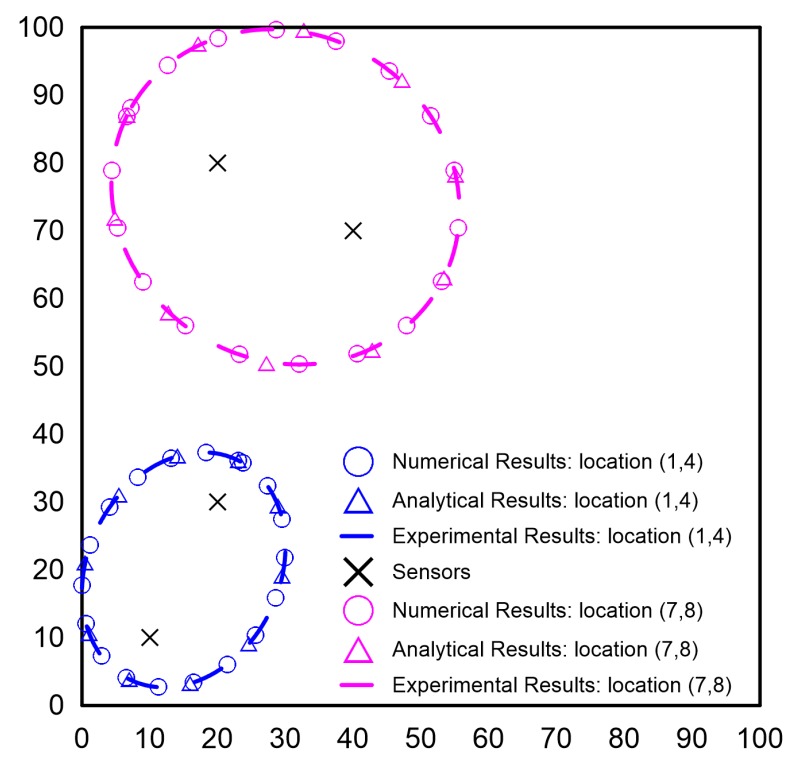
Validation of the analytical approach [11].

**Figure 12 sensors-19-01856-f012:**
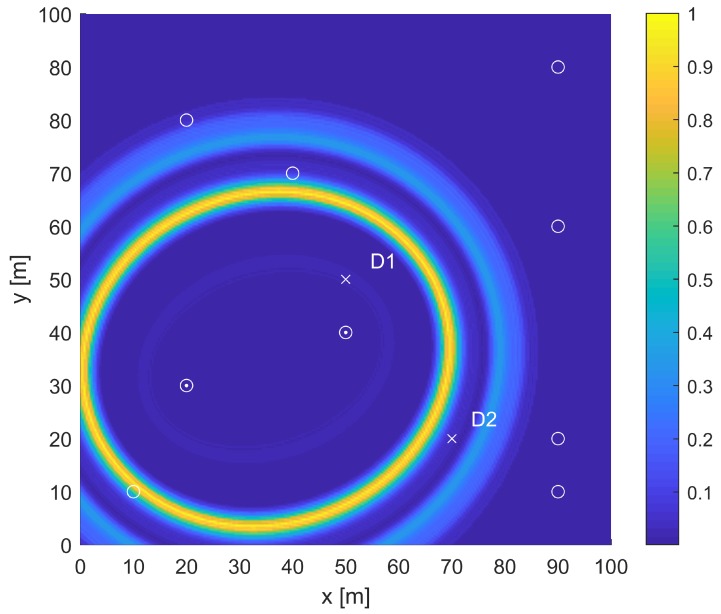
Ellipses based on two sensor locations using SLDV measurements with excitation at Sensor 5 and sensing at the measurement point corresponding to Sensor 4 location; excitation: 200-kHz tone burst; calculations using wave velocity (5300 m/s) corresponding to S0 mode [30].

**Figure 13 sensors-19-01856-f013:**
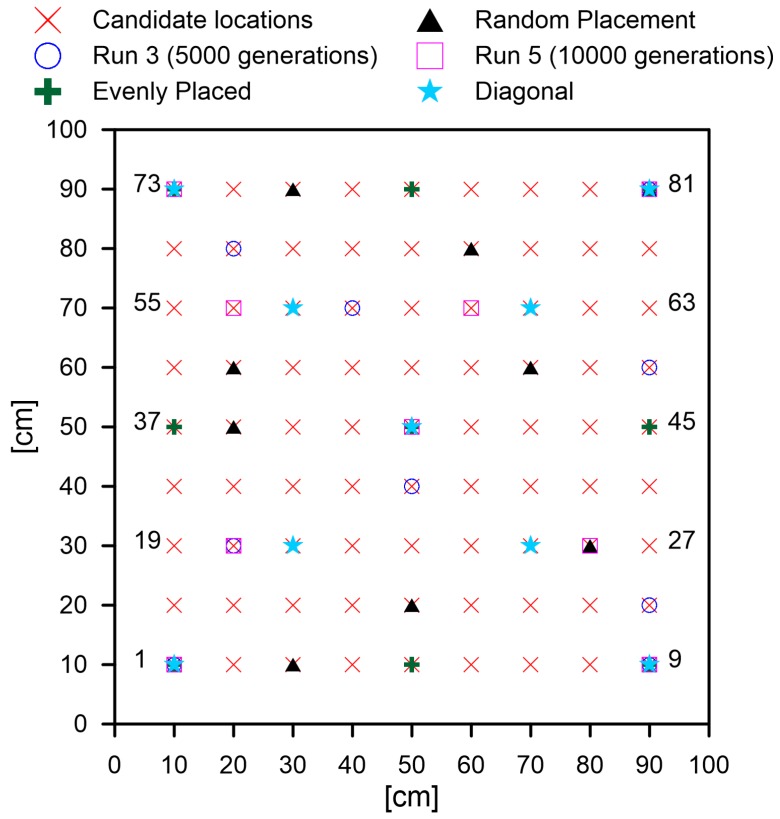
Nomenclature of possible sensor locations and sensor placement for different runs.

**Figure 14 sensors-19-01856-f014:**
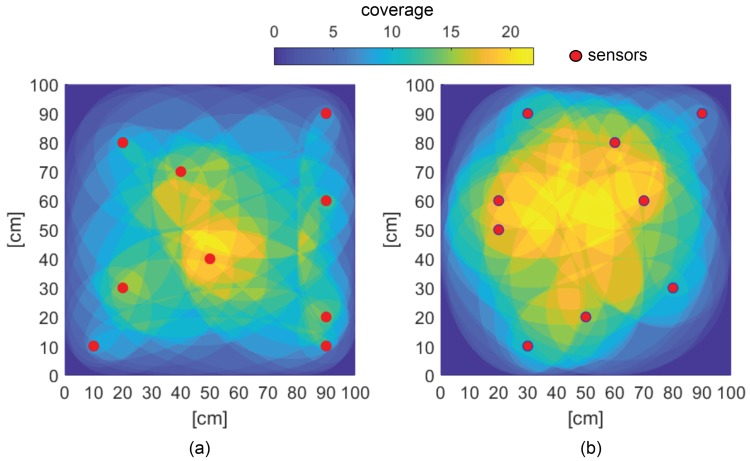
Surface plot showing coverage: (**a**) Optimized placement; (**b**) Random placement.

**Figure 15 sensors-19-01856-f015:**
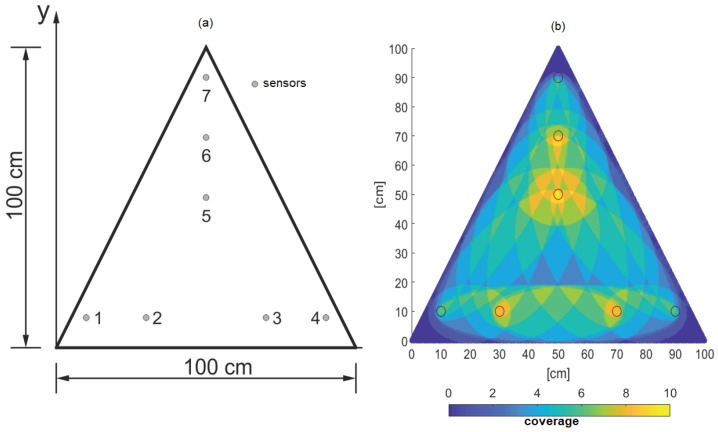
(**a**) Geometry and arrangement of piezoelectric transducers in the triangular aluminum plate; (**b**) Surface plot showing coverage for the triangular plate.

**Figure 16 sensors-19-01856-f016:**
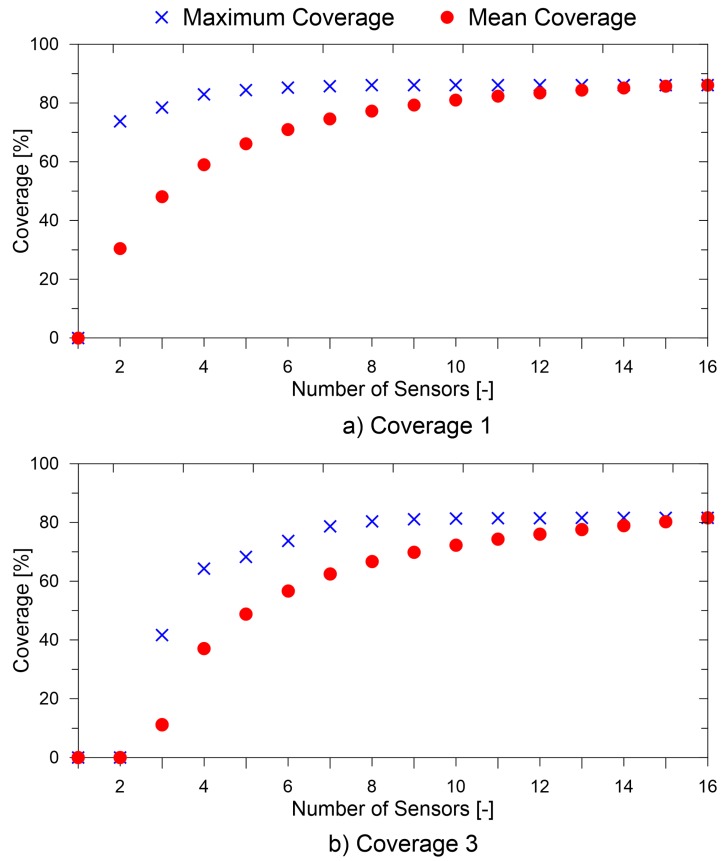
(**a**) Relation of objective Coverage1 with the number of sensors; (**b**) Relation of objective Coverage3 with the number of sensors.

**Figure 17 sensors-19-01856-f017:**
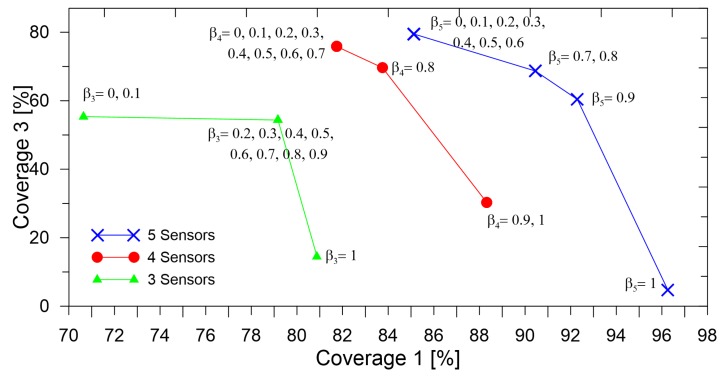
Pareto front for different values of β for different numbers of sensors.

**Table 1 sensors-19-01856-t001:** Performance parameters for different sensor placements.

Run	Generations	Placement	Number	coverage1	coverage3
1	5000	1, 9, 10, 21, 36, 40, 44, 46, 73, 81	10	97.41%	90.77%
2	5000	1, 9, 11, 19, 25, 52, 53, 64, 66, 69, 74, 81	12	97.78%	94.51%
3	5000	1, 9, 18, 20, 32, 54, 58, 65, 81	9	96.10%	88.66%
4	5000	1, 9, 21, 23, 67, 69, 72, 73	8	96.68%	87.07%
5	10,000	1, 9, 20, 26, 40, 56, 60, 73	8	96.65%	90.55%
even	-	1, 5, 9, 37, 41, 45, 73, 77, 81	9	95.58%	90.25%
diagonal	-	1, 9, 21, 25, 41, 57, 61, 73, 81	9	97.72%	94.85%
random	-	3, 14, 26, 38, 47, 52, 69, 75, 81	9	90.61%	86.43%

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
