# Peer review of "A Study of Sensor Placement Optimization Problem for Guided Wave-Based Damage Detection†"

_sensors, 2019, doi:10.3390/s19081856_

Author Response

Dear Reviewer,

We would like to thank for the valuable comments to our manuscript. Below are the cited comments (in black) with our answers (colour coded for different reviewers). All the changes in the manuscript are colour coded with similar colours for each reviewer.

I would almost recommend its publication as it is. However, I have some comments I would suggest to the authors to incorporate into the text to enhance the overall strength of the paper. For instance, I suggest to increase the bibliography with some works that in my opinion may help the reader:

The authors will like to thank the reviewer for their positive assessment of the manuscript. The point by point response and corrections made for each of the raised points below are given here.

1)    Giurgiutiu, V., Soutis, C. Guided Wave Methods for Structural Health Monitoring, 2010;

2)    J. L. Rose, Ultrasonic Waves in Solid Media, Cambridge University Press., 1999.

Indeed, I think that these books may complete the actual ref. 19.

Thank you for the relevant remark. Indeed these works are seminal in the field of GW. The references were added to the manuscript for the general readers.

3) Pages 1-2, lines 30-32. When talking about "GW sensitiveness to small levels of

damage" and "originating from different mechanisms including impacts" I suggest to cite: "Metamaterials-based sensor to detect and locate nonlinear elastic sources, AS Gliozzi et al., Applied Physics Letters 107 (16), 161902 - 2015" and "Proof of concept for an ultrasensitive technique to detect and localize sources of elastic nonlinearity using phononic crystals, M Miniaci et al., Physical review letters 118 (21), 214301 -2017".

When talking about signal processing, authors may also provide the following

references to further show the difficulties of dealing with dispersive signals: "A passive monitoring technique based on dispersion compensation to locate impacts in plate-like structures De Marchi et al., Smart Materials and Structures 20 (3), 035021" and "De Marchi et al., A dispersion compensation procedure to extend pulse-echo defects location to irregular waveguides, 2013 NDT & E International 54:115-122".

The reviewers agree with the comment and the references were included at appropriate locations in the literature review.

Minor comments I may recommend to improve the work are:

I suggest to uniform the font size in the figures (see for instance fig. 5)

Thank you for noticing the error. The figure has been improved and updated

Fig. 6. I would use larger font for the numbers and I would choose another color to

enhance the contrast (white or red for instance).

Thank you for the suggestion, the figure has been improved for better contrast and higher visibility

I would extend the captions of some figures (for instance, fig. 9 has a very nice self explicative caption.

Thank you for the advice. The captions of some of the figures have been edited for clarity.

Reviewer 2 Report

The authors aim at optimizing the sensor placement at for SHM of plate like structures with guided elastic waves.

While the paper is in good English some central ideas are given only implicitly, and the reader has to go over the paper several times to grap some main ideas.

In the following has to be considered only as examples.

In optimization you write (line 92):

The application requirements are maximum coverage of the sensor network by at least one sensor-actuator pair which ensure damage detection.

For damage isolation coverage with 3 sensor-actuator pairs is necessary. Each pair defines an ellipse, so having three ellipses their intersection pinpoints the damage location.

How you define coverage? This is given somewhere else. Your criterion is the area of an ellipse with the two sensor points centres.

Why do you use the largest ellipses within the plate as criterion for your coverage1 and coverage3 calculations? Please give more motivation and clear explanation. It would be good to start from

Another example: How is the procedure to get the points in Figure 8? The chapter 3.2 (experimental approach) says nothing how you get the experimental points!  You even do not mention there, that you do apply Laser Doppler vibrometry. The reader experiences, what is done experimentally only in Figure 9, and only after heavy guessing.

I recommend to restructure the manuscript and to make it more precise and clear.

Author Response

Dear Reviewer,

We would like to thank for the valuable comments to our manuscript. Below are the cited comments (in black) with our answers (colour coded for different reviewers). All the changes in the manuscript are colour coded with similar colours for each reviewer.

The authors aim at optimizing the sensor placement at for SHM of plate like structures with guided elastic waves.

While the paper is in good English some central ideas are given only implicitly, and the reader has to go over the paper several times to grap some main ideas.

The authors will like to thank the reviewer for their positive assessment of the manuscript. The point by point response and corrections made for each of the raised points below are given here. We hope that the corrections are to the expectation of the reviewer

In the following has to be considered only as examples.

In optimization you write (line 92):

The application requirements are maximum coverage of the sensor network by at least one sensor-actuator pair which ensure damage detection.

For damage isolation coverage with 3 sensor-actuator pairs is necessary. Each pair defines an ellipse, so having three ellipses their intersection pinpoints the damage location.

How you define coverage? This is given somewhere else. Your criterion is the area of an ellipse with the two sensor points centres.

The coverage0 is when the location is not covered by any ellipse, coverage1 is when the location is covered by 1 ellipse originating from 1-sensor-actuator pair, coverage3 where the location is covered by at least 3 ellipses originating from at least 3 sensor-actuator pairs.

The text has been edited for clarity. An additional plot has been added to improve the understanding of the terminology used.

Why do you use the largest ellipses within the plate as criterion for your coverage1 and coverage3 calculations? Please give more motivation and clear explanation. It would be good to start from

The largest ellipse which is inscribed in the plate is the largest area covered by 1 sensor-actuator pair without facing problems for the scatter from the boundary.

A discussion about the use of ellipses as well as the decision of using the smallest ellipse has been added. In addition the plot for the determination of the ellipses has been added to improve the understanding.

Another example: How is the procedure to get the points in Figure 8? The chapter 3.2 (experimental approach) says nothing how you get the experimental points! 

The authors agree that the procedure needs to be provided.

A figure showing the flowchart for the transformation for the A-scan to the ellipse has been added to the manuscript.

 You even do not mention there, that you do apply Laser Doppler vibrometry. The reader experiences, what is done experimentally only in Figure 9, and only after heavy guessing.

The description of the equipment was provided along with the motivation for the use of sensors and LDV for the measurements.

I recommend to restructure the manuscript and to make it more precise and clear.

The manuscript has been considerably edited with new figures and discussions to minimize the guessing on the part of the reader.

Reviewer 3 Report

In this paper, an optimization method of sensor placement is proposed for damage detection and isolation using guided wave. The genetic algorithm has the advantages of global optimization, but it takes relative large computation to get the optimal point. How to choose cost function is also important.  Some discussion can be added to compare the genetic algorithm with other traditional optimization method regarding sensor placement. Will the proposed approach have the potential to be applied to the health monitoring of composite plates?  In the conclusion section, it is suggested to emphasize the significance of the proposed approach in the industrial application. 

Author Response

Dear Reviewer,

We would like to thank for the valuable comments to our manuscript. Below are the cited comments (in black) with our answers (colour coded for different reviewers). All the changes in the manuscript are colour coded with similar colours for each reviewer.

In this paper, an optimization method of sensor placement is proposed for damage detection and isolation using guided wave.

The authors will like thank the reviewer for his time and the critical comments which have allowed us to improve the manuscript. The pointwise response to each of the concern is given below and appropriate changes have been made in the manuscript in orange color.

The genetic algorithm has the advantages of global optimization, but it takes relative large computation to get the optimal point. How to choose cost function is also important.  Some discussion can be added to compare the genetic algorithm with other traditional optimization method regarding sensor placement.

The authors agree that the computational demands for the GA are quite large, but that is true for almost all the meta-heuristic methods. But their ability to effectively search the sample space for the optimal is better than the deterministic optimization algorithms such as the sequential placement. Also the reviewer is correct in pointing out that the choice of cost function can determine the success of the optimization process. Hence discussions are included in the manuscript about the choice of the cost function. Also a discussion on why the comparative study is not conducted has been included in the paper on page 4 and page 5.

Will the proposed approach have the potential to be applied to the health monitoring of composite plates?

Yes the proposed approach can indeed be applied for composite plates by using the directional velocity of the wave for constructing different ellipses for all possible damage locations. The statement has been supported with appropriate references. In addition other approach using ray tracing approach tried by researchers has been discussed as well. In spite of the choice of the ray-tracing or the ellipse approach the other key steps of the optimization remain unchanged and hence the methodology has merit and at the same time wide applicability.

In the conclusion section, it is suggested to emphasize the significance of the proposed approach in the industrial application. 

A short passage has been added identifying what are the future steps which need to be taken before the methodology is used for industrial application. The use of the methodology for real structures is envisaged as a future direction of research

Reviewer 4 Report

This paper is about a study of sensor placements optimization for SHM with guided waves. The authors present a complex scheme for choosing optimal transducer positions. However they fail to give a clear-cut description of their methodology. A set of flowcharts is presented without giving clear connection to the present experimental setup.  A-scans (amplitude versus time) are transformed into ellipses on the examined plate without giving the transformation scheme. The outcome of the optimization process remains unclear. Which is the optimum selection of the transducers and why? No damage in the plate is included.

Given all these negative points the reviewer proposes the rejection of the manuscript for publication. There are too many unexplained topics.

Some further details:

p 4, eq 1: The meaning of the quantities coverage1 and coverage3 must be described more clearly

p 4, Numerical approach: the numerical simulation of the wave propagation is not sufficiently described. What software was used?

p 6, Experimental approach: give type and manufacturer of the used equipment (pulser, receiver, transducer, digitizer)

p 6, Fig 6: give information about the grid (10 cm x 10 cm ?). Why was this grid used and not another?

p 7, eqs 2 and 3: add a sketch of the ellipse which contains all relevant elements given in these equations, especially: alpha, lcx, tangent line, M. Describe in detail the meaning of this ellipse and the relation to the measured A-scan.

p 7, line 196: the sampling rate is not given. Looking at Fig. 7 there is no evidence of undersampling at the blue curve. It is unclear what is the cause of the signal at about 25 microsec.

p 8, Fig. 8: It remains unclear what signals are plotted as the ellipses. For a given pair of transducers there is one single A-scan, as shown in Fig. 7. How is this A-scan transformed into an ellipse as shown in Fig. 8?

p 8, line 203: give type and manufacturer of the laser vibrometer. It is unclear why a laser vibrometer was used. Why not just taking the signal of the receiver transducer?

p 8, Tab. 1: give clear-cut definition of Generations. It is unclear what is iterated in the different runs. Again, the real meaning of coverage1 and coverage3 remains unclear.

p 9, Fig. 9: Again: the laser vibrometer measures an A-scan. How is this A-scan transformed into the 2d plot in Fig. 9?

p 10, top: the authors fail to show a) what quantity is optimized and b) how does the optimized configuration look like.

p 10, line 222: how is this value calculated and what is the meaning of it?

p 10, Fig. 11: It is not described what quantity is displayed

p 10, line 247: how is this value calculated?

p 10, line 251: it remains unclear what is meant with chromosomes, mutation rate and elitism in the present context.

p 11, Fig. 11: it is unclear why this configuration is the optimum

Author Response

Please find the response in the attached file

Round  2

Reviewer 2 Report

the manuscript has improved considerably. I have no further objection to the publication.